# Galectins: Important Regulators in Normal and Pathologic Pregnancies

**DOI:** 10.3390/ijms231710110

**Published:** 2022-09-03

**Authors:** Min Chen, Jia-Lu Shi, Zi-Meng Zheng, Zhi Lin, Ming-Qing Li, Jun Shao

**Affiliations:** 1Laboratory for Reproductive Immunology, Hospital of Obstetrics and Gynecology, Shanghai Medical School, Fudan University, Shanghai 200080, China; 2NHC Key Lab of Reproduction Regulation, Shanghai Institute for Biomedical and Pharmaceutical Technologies, Fudan University, Shanghai 201203, China; 3Shanghai Key Laboratory of Female Reproductive Endocrine Related Diseases, Hospital of Obstetrics and Gynecology, Fudan University, Shanghai 200080, China; 4Department of Obstetrics, Hospital of Obstetrics and Gynecology, Shanghai Medical School, Fudan University, Shanghai 200011, China

**Keywords:** galectin, maternal–fetal interface, pathologic pregnancy, preeclampsia, gestational diabetes mellitus, fetal growth restriction, preterm birth

## Abstract

Galectins (Gal) are characterized by their affinity for galactoside structures on glycoconjugates. This relationship is mediated by carbohydrate recognition domains, which are multifunctional regulators of basic cellular biological processes with high structural similarity among family members. They participate in both innate and adaptive immune responses, as well as in reproductive immunology. Recently, the discovery that galectins are highly expressed at the maternal–fetal interface has garnerd the interest of experts in human reproduction. Galectins are involved in a variety of functions such as maternal–fetal immune tolerance, angiogenesis, trophoblast invasion and placental development and are considered to be important mediators of successful embryo implantation and during pregnancy. Dysregulation of these galectins is associated with abnormal and pathological pregnancies (e.g., preeclampsia, gestational diabetes mellitus, fetal growth restriction, preterm birth). Our work reviews the regulatory mechanisms of galectins in normal and pathological pregnancies and has implications for clinicians in the prevention, diagnosis and treatment of pregnancy-related diseases.

## 1. Introduction

The maternal–fetal interface features a number of complex and tightly regulated mechanisms. These include physical defenses through intercellular junctions, the secretion of cytokines, and protection against microbial infections through the innate immune response. In addition, the interaction between fetal trophoblast and maternal decidual immune cells can also promote immune tolerance [1,2]. Studies suggest that a successful pregnancy is the result of multiple steps, including maternal immunological adaptation, normal blastocyst development, functional placental development and endometrial receptivity formation [3]. During early gestation, trophoblasts in the human placenta have two main differentiation pathways: they differentiate into chorionic villous trophoblasts and extrachorionic villous trophoblasts. Extravillous trophoblasts (EVT) have both migratory and invasive phenotypes for remodeling spiral arteries and can invade the uterus [4].

Galectins are believed to play a crucial role in reproductive processes, such as maternal–fetal immune tolerance, embryo implantation and angiogenesis [5]. Evidence points out that galectins are involved in the establishment and maintenance of normal pregnancy. Gal-1, gal-3 and gal-9 are the main participants in these processes, with other family members also contributing significantly to immune–endocrine interactions and maternal–fetal immunological responses [5,6]. Galectins show diverse intra and extracellular localization and biological functions [7]. Through a number of clinical trials, scientists have found that their most considerable role in pregnancy appears to be the modulation of the maternal–fetal immune response, with some galectins suppressing the mother’s immune response to the fetus, thus maintaining a normal pregnancy [8]. Therefore, we reviewed the role of galectins in normal and pathological pregnancies.

## 2. The Galectins (Gal) Family

Galectins are a family of galactoside-binding proteins found in animals, bacteria and fungi [9]. They consist of a core sequence of 130 amino acids and a highly conserved carbohydrate recognition domain (CRD) [10]. Two defining characteristics of the lectin family including: significant similarity in the amino acid sequence and high affinity for galactoside [11]. Currently, researchers have found 15 galectins in mammals, including gal-1-15. On the basis of their structural differences, these galectins have been categorized into three types (Figure 1): prototype galectin, chimeric galectin and tandem repeat galectin. Prototype galectins contain a single carbohydrate recognition domain, including gal-1, gal-2, gal-5, gal-7, gal-10, gal-11, gal-13, gal-14 and gal-15. Chimeric galectins are self-associated with a c-terminal CRD and a non-carbohydrate bound n-terminal structural domain, while gal-3 is the only member of this group. Tandem repeat galectins are dimers consisting of a linker peptide joining two CRDs, and these include gal-4, gal-6, gal-8, gal-9 and gal-12 [12,13].

In a large number of different cell types, galectins are found both intracellularly, such as in the nucleus, cytoplasm and cell membrane, and extracellularly, especially in the extracellular matrix [14]. Galectins are synthesized in the cytosol on free ribosomes [15,16]. Because they lack a signal sequence for secretion, which is probably related to their ability to bind glycans, they utilize nonclassical secretion pathways [14]. As a result, vesicle-disruption-induced accumulation and binding to luminal glycans may serve as a molecular link between autophagy and galectin secretion [17]. However, in another study, the maturation of N-linked glycoprotein was not necessary for galectin-3 transport from the cytosol to the extracellular space, and the majority of the released galectin-3 was not packaged into the extracellular vesicles [18]. The results of a study on galectin-3 showed that oligomerization may be required for galectins secretion [19,20,21].

Galectin-1 may secrete similarly to fibroblast growth factor-2 (FGF-2), the most researched unconventionally secreted protein that follows the direct translocation pathway [22]. Galecitins are released from cells in the extracellular vesicles (EVs), either in the microvesicles created by membrane blebbing or in the exosomes derived from multivesicular bodies (MVBs) [23]. Extracellular matrix components and some inflammatory factors can affect the secretion of galectins [13,24]. Extracellular galectins are capable of binding to various cell surface receptors to form carbohydrates [25]. Tyrosine kinase receptors and T cell glycoproteins (TCR, CD3, CD43, CD45 and CD7) are just two of the many receptors that galectins interact with to control cytokine signaling and receptor activation. Integrins and galectins collaborate to control cell adhesion and death. Additionally, contact with Gal3 encourages lamellipodia generation, actin and focal adhesion turnover and fibronectin modification all increase cell adhesion and migration [26]. Additionally, recombinant galectins can exert multiple in vitro activities by binding cell surface glycans and the extracellular matrix [27].

## 3. The Expression of Galectins at Maternal-Fetal Interface

The maternal–fetal interface connects maternal tissue and fetal components in direct contact, and its local immune response plays a role in protecting the fetus during the establishment and maintenance of pregnancy and the onset of labor [28]. Gal1-4, gal-9, and gal-12 are expressed in the endometrium, with gal-1 and gal-3 expressing the most. Immunohistochemistry detected gal-1 in the endometrial stroma, as well as the decidua and gal-3 in the endometrial glandular epithelium [29]. This was also demonstrated by immunohistochemistry, in post-pregnancy mice, where the gal-3 was located mainly in the luminal epithelium and glandular epithelium and reached a maximum at 2–4 and 6–8 days, respectively [30]. Gal-3 was rare in non-pregnant animals or during preimplantation. Later in pregnancy, gal-3 was found in the decidual basement, uterine glands and placental trophoblast cells. Decidualized and non-decidualized endometrium lacked gal-3 [31]. Finally, gal-3 was shown to be associated with endometrial receptivity and implantation [30,32].

Galectins are abundantly expressed in the reproductive system, and their expression in the human reproductive system is summarized in Table 1. During the first trimester, gal-1 is mainly expressed in human placental cytotrophoblasts (CTB) and syncytiotrophoblast (STB) [33,34]. Gal-1 also was detected in decidual cells, suggesting that this galectin promotes inter-trophoblast and trophoblast–stromal cell interactions during placental formation interactions [35]. Additionally, the double immunofluorescence confirmed that the expression of gal-2 was mainly in syncytiotrophoblast and maternal decidua [36] but also in extravillous trophoblast and fetal endothelial cells [37]. Gal-2 is primarily immunomodulatory through its anti-inflammatory effects [38].

Gal-3 was found by immunohistochemistry, which is expressed in all trophoblastic lineages but decreased in the villous trophoblast (VT) and trophoblast columns in first and third trimester [35,37], suggesting that gal-3 expression is associated with a shift in cell phenotype, namely, the change from a proliferative to a migratory phenotype [39]. As pregnancy progresses, circulating levels of gal-3 gradually increase, indicating that its expression is regulated throughout development [39,40]. Through in vitro experiments, investigators found that exogenous gal-3 positively regulates trophoblast function and induces cell invasion, tube formation and fusion [41]. Gal-7 was immunolocalized to syncytiotrophoblast, extravillous cytotrophoblast and glandular epithelial cells of the placenta in early pregnancy and to syncytiotrophoblast and endothelial cells in term placentas, but no endothelial cell staining was seen in pre-eclamptic placentas [42]. Results by immunolocalization showed gal-8 expression in both villi and EVT [43].

Gal-9 mRNA was expressed in the human endometrium, especially in endometrial epithelial cells, and was significantly increased in the mid-and to late-secretory stages, in the window of implantation and in the decidua [44]. Ultrastructural immunocytochemistry confrimed the localization of gal-9 in the endometrium was, where it was mainly located in the apical protrusions of the endometrial epithelium, a type of protrusion also known as the uterodome [44]. It plays a role in endometrial receptivity; however, there was no gal-9 observed between the uterodomes [45]. Importantly, compared with other kinds of T cells, gal-1 and -10 have much higher expression levels in CD4+ CD25+ Treg cells, where they perform an essential role in the suppression of immune responses [46,47].

Gal-12 is minimally expressed in adipocytes and is required for in vitro adipocyte differentiation. This protein can regulate in vivo lipolysis, total body adiposity, and glucose homeostasis [48]. Gal-13 (placental protein 13, PP13) was found in STB in chorionic villi and sometimes in multinucleated luminal trophoblasts within converted decidual spiral arterioles. However, gal-13 was not detected in the cytotrophoblast, and the anchoring trophoblast and invasive trophoblast [49,50]. In early gestation, gal-14 is predominantly expressed in the STB and its placental expression is decreased in women with miscarriage [7]. LGALS15 (a gene of gal-15) is only expressed in Caprinae endometrium and serves as a peri-implantation attachment factor [51].

## 4. The Role of Galectins in the Maternal–Fetal Interface

### 4.1. Immune Regulation

Successful pregnancy is a complex physiological process and a major immune challenge [55]. It requires the synchronization of endometrial and embryonic development and maintenance of a delicate equilibrium between inflammation and immune tolerance [56]. The endometrium undergoes decidualization in response to the regulation of pregnancy-related hormones, accompanied by the enrichment and reissue of immune cell subpopulations [57,58]. A normal pregnancy is similar to a successful pure heterozygous semi-allogeneic transfer in which the mother does not reject the embryo carrying the father’s antigens, but rather establishes a unique immune tolerance mechanism through a subtle immune dialogue [28,59,60]. The maternal–fetal interface consists mainly of trophoblast cells, decidua stromal cells and decidua immune cells, which, under hormonal regulation, produce a variety of cytokines that create a specific immune tolerance environment between the mother and the fetus, facilitating a successful pregnancy [3,61].

Additionally, the regulation of multiple galectins is involved in regulating maternal–fetal immune tolerance [62,63]. Almost all immune cells can express galectins, which are upregulated in activated B cells, T cells, macrophages and decidual natural killer (dNK) cells [63,64]. Thus, galectins make a difference in maternal–fetal immunotolerance (Figure 2).

During pregnancy, gal-1 modulates the inflammatory response, promotes immune tolerance and prevents maternal rejection of the fetus. Therefore, it is a key regulator of maternal–fetal immune tolerance and may have therapeutic implications for pathologic pregnancies [63]. This is because recombinant gal-1 induces tolerogenic dendritic cells, promotes the secretion of interleukin-10 (IL-10) and regulates the expansion of T cells. In addition, gal-1 and progesterone have a synergistic effect in maintaining pregnancy [63].

Gal-1 has pro-apoptotic activity on activated CD4+ T cells of the subtypes Th1, Th17 and CD8+ T cells. dNK cells can produce gal-1 [53], and the supernatant of cultured dNK cells can induce apoptosis in T cells, which can be blocked by anti-gal-1 antibodies. Instead of peripheral T cells, decidual T cells can bind gal-1. This suggests that decidual immune cells form a privileged maternal–fetal immune microenvironment by secreting gal-1 and binding gal-1 [65]. Molvarec et al. found that gal-1 is possibly related to innate and acquired immune cell activation [66].

Gal-1 is regulated by estrogen, which may be one of the regulatory mechanisms involved in maternal–fetal immune tolerance [67]. The estrogen-responsive element in the promoter of LGALS1 (gene of gal-1) is conserved human cis motifs, present in the placenta and involved in how steroid hormones influence the level of gal-1 expression. Amino acid substitutions occur at key residues in early mammalian evolution, including the acquisition of cysteine residues, which regulate immune function through redox-state-mediated conformational changes that disable sugar binding and dimerization [68]. In addition, gal-1 regulates the expression of human leukocyte antigen G (HLA-G) on EVT cells, thereby promoting maternal–fetal immune tolerance [69]. These are the possible mechanisms by which hormonal and redox responses regulate the involvement of the gal-1 in immune responses.

Gal-3 is essential during gestation, and in mice it is mainly located at the embryo implantation site. The binding partner of gal-3, cubilin, was isolated in the uteroplacental complex. In the last week of pregnancy in mice, cubilin co-localized with gal-3 in the yolk sac epithelium and was found in uterine NK cells [70]. Furthermore, cytoplasmic gal-3 protects T cells from apoptosis while increasing cell proliferation [71]. In contrast, gal-3 induced T cell apoptosis [72] and inhibited CD-66a expression [73].

Gal-9 expression is increased during pregnancy [74], which was expressed in the placental spongiotrophoblast, where decidual immune cells displayed lower toxicity and higher PD-L1 expression levels relative to peripheral immune cells. T-cell immunoglobulin and mucin domain 3 (Tim-3) plays a function in immune control by attaching to its ligand, gal-9, which triggers effector T cell exhaustion or apoptosis [75]. Local Tim-3 expression was higher than in the periphery and decidua, which reduced lysis activity. In preeclampsia, maternal immune cells (T cells, cytotoxic T cells, NK cells, CD56(dim) NK cells) express less Tim-3 [76,77]. Tim-3/Gal-9 decreases NK cell toxicity to trophoblast cells by converting NK cells into dNK [78]. Regulatory T cells (Tregs) produce gal-9 at increasing levels to maintain maternal–fetal immunological tolerance as pregnancy progresses [79]. He et al. suggest that gal-9 can modify PBMC function to Th2 bias, maintaining pregnancy [80]. IL-27 and gal-9 can synergistically induce Tim-3+Treg cells in vitro [81]. Tim-3(+) pNK cells release anti-inflammatory cytokines and activate regulatory T cells in a TGF-1-dependent manner, causing immunosuppression. Tim-3(+) pNK cells decreased miscarriage risk in NK-deficient mice. Moreover, Tim-3/Gal-9 signaling regulates immunological control by pNK cells, a critical player in maternal–fetal immune tolerance [82].

The Tim-3/Gal-9 pathway stimulation causes midterm M2 macrophage conversion. Tim-3 and gal-9 expression were elevated in dysfunctional decidual macrophages at embryonic day 9(E9), showing that this pathway is engaged in early pregnancy and embryo development [83]. The LGALS9 D5 isoform inhibits interferon production by decidual natural killer cells. In human spontaneous abortion patients, researchers detected six LGALS9 splice variants [84]. Tim-3/Gal-9 modulates the cellular activity of dNK to maintain normal pregnancy and a result has been confirmed in human aborted decidua and in a mice miscarriage model with a reduced percentage of Tim(+) dNK cells [85].

Gal-10, also known as Charcot–Leyden crystal protein, is the most abundant protein in eosinophils and can form characteristic crystals in tissues and secretions at sites of eosinophil-associated inflammation. Gal-10 is a characteristic expression feature of suppressor T cells, eosinophils and basophils [46,86,87].

Than et al. demonstrated that placenta-specific galectins (such as gal-13 and gal-14) are predominantly expressed by STB and induce apoptosis in T lymphocytes [64]. Gal-13 and gal-14 can induce apoptosis in Th and Tc cells. Examining activation markers revealed that gal-13 increased CD25 expression and gal-14 decreased CD71 expression on the cell surface, while both galectins increased CD95 expression on T cells [7]. In the presence of gal-13 and gal-14, inactivated T cells were capable of producing substantial amounts of IL-8. These are the mechanisms involved in the regulation of maternal–fetal immunity by gal-13 and gal-14 [7]. By decreasing the rate of apoptosis, gal-13 appears to make a significant contribution to the control of placental neutrophils by raising the production of PD-L1, HGF, TNF-α, ROS, and MMP-9, therefore polarizing them toward a placental-growth-permissive phenotype [88]. In addition, gal-13 aggregates in the decidua may act as decoys to induce apoptosis and promote maternal immune tolerance to pregnancy [89].

### 4.2. Embryo Implantation

Successful embryo implantation is a complicated procedure, which requires the embryo and the endometrium to work together in order to coordinate a sequence of events that occur during the procedure [90,91]. An intricate chemical chain reaction, which is controlled by endocrine, paracrine and autocrine regulators present in both the embryo and the mother, is essential to the progression of embryo implantation [92]. The results of in vitro and in vivo studies have demonstrated that galectins are crucial mediators in the implantation process [93]. During embryo implantation, the enhanced expression of gal-1, -3 and -9 in endometrial epithelial cells reflect the galectins’ most crucial involvement in endometrial receptivity. This is due to the increased expression of these genes in endometrial epithelial cells [29,45].

Gal-1 is an important downstream target of the P(4)-FKBP52-PR signaling pathway in the uterus, which enhances P(4) responsiveness during pregnancy, and the activation of this pathway reduced the rate of mid-gestation resorption in mice and rescued implantation failure [94]. In the absence of embryos, gal-1 expression decreases. In delayed implantation mice, gal-1 mRNA levels increase with the termination of the implantation delay [67]. Progesterone and estrogen oppose each other in uterine gal-1 mRNA levels. RU486 (progesterone receptor antagonist) reduced progesterone-induced gal-1 mRNA in ovariectomized mice uterine tissues. This regulation corresponded with the implantation procedure [67].

The discovery of complex connections between gal-3 and integrin β3 in the control of endometrial cell proliferation and adhesion provides an in vitro model applicable to embryo implantation and endometrial receptivity [95]. BeWo cells are stimulated to produce and secrete gal-3 by 17-estradiol (E2), progesterone and human chorionic gonadotropin (hCG). Through activation of integrin β1, recombinant gal-3 inhibited endometrial cell (RL95-2 cells) proliferation and induced apoptosis. In vitro tests confirmed the pro-apoptotic action of trophoblast-secreted gal-3 on endometrial cells [96]. Gal-3 has an anti-apoptotic impact on endometrial cells, and estrogen and progesterone can modify gal-3 synthesis [97]. In animal trials, when the gal-3 gene was knocked out in mice, considerably fewer embryos were implanted in the mice endometrium. In conclusion, embryo implantation requires an increase in gal-3 expression after pregnancy [30]. In contrast, researchers demonstrated that mice can undergo embryo implantation even without gal-1 and gal-3. Additionally, gal-5 is present during the window of implantation in the blastocyst [98].

Gal-7 is a potentially useful blood biomarker for preeclampsia and may play a significant part in the implantation of the placenta [42]. In addition, gal-7 expression is elevated in the endometrial epithelium and stroma of women with a history of miscarriage. The findings imply that gal-7 promotes the embryo’s adhesion to the endometrium and that higher gal-7 levels may cause pathological pregnancy [54]. It is expressed at low levels throughout the proliferative and early secretory phases and sharply increases during the mid- and late-secretory phases, the window of implantation and in the decidua [44], indicating that gal-9 may have been involved in embryo implantation.

### 4.3. Angiogenesis

A number of vascular processes that need to be coordinated in a spatial and temporal manner at the interface between the mother and the fetus for pregnancy to be successful [99,100]. In the early stages, the embryo is able to successfully implant in a vascularized and receptive uterus thanks to the hormone-mediated modification of the endometrial vascular system [101,102]. In order to guarantee that the embryo is provided with oxygen and nutrients prior to the formation of a definitive placenta, this is accompanied by vasodilation and the formation of neonatal structures during the decidualization process [103]. The vascular system of the placenta continues to be remodeled as the pregnancy progresses reach needs of the fetus [103]. Disturbances in these processes are frequently associated with unfavorable pregnancy outcomes such as preeclampsia, intrauterine growth restriction (IUGR) or preterm birth [104].

The result of oncology studies have found that gal-1 can regulate tumor angiogenesis and can be a potential therapeutic target to reduce angiogenesis [105]. Researchers discovered that gal-1 demonstrates a pro-angiogenic function in early gestation, promoting vasodilation through vascular endothelial-derived growth factor (VEGF) receptor 2 signaling [106]. Gal-1 may be involved in mechanisms related to placental and maternal spiral artery remodeling. Gal-1 deficiency manifests as spontaneous PE-like disease in mice [106]. Through its mRNA-binding function, gal-1 is able to regulate angiogenesis by binding to the mRNAs of genes that are linked with angiogenesis [107]. OTX008, an inhibitor of gal-1, was found to inhibit tumor proliferation, invasion and angiogenesis [108].

The result of endometriosis studies have confirmed the involvement of gal-1 in angiogenesis. Using experimental endometriosis models induced in wild-type and gal-1-deficient (LGALS1(−/−)) mice, researchers demonstrated that gal-1 regulates the formation of vascular networks in endometriotic lesions and contributes to the growth of their ectopic foci, independently of VEGF and plasmacytoid-derived CXC-motif (CXC-KC) chemokines [109]. An increase in angiogenesis can be attributed to the synergistic impact of gal-1 and gal-3, which works by activating VEGFR-1. This activation of VEGFR (vascular endothelial-derived growth factor receptor)-1 may be related to a decrease in receptor endocytosis [110,111]. The absence of gal-3 or its inhibition led to a significant reduction in the implantation and size of endometriotic lesions, as well as the expression of VEGF and VEGFR-2, and the vascular density [112].

Gal-2 expression was significantly lower in decidua and extravillous cytotrophoblast, possibly due to its role in angiogenesis [113]. In tumor patients, an increased circulation of gal-2, -4 and -8 interacts with the vascular endothelium and significantly promotes the increased circulation of granulocyte colony-stimulating factor (G-CSF), IL-6 and monocyte chemoattractant protein-1 (MCP-1). In turn, these cytokines and chemokines boost the activity of endothelial cells during angiogenesis and metastasis [114]. Only gal-2 expression was dramatically reduced in spontaneous abortion and recurrent abortion (RA) placental trophoblast cells [113]. In studies of patients with coronary artery disease, research has confirmed that gal-2 is a novel inhibitor of arteriogenesis. The modulation of gal-2 may become a new therapeutic strategy for stimulating arteriogenesis in patients with coronary artery disease [115].

Troncoso et al. describe a unique role for gal-8 in the regulation of vascular and lymphatic angiogenesis and give evidence of its critical impacts on tumor progression [116]. Gal-8 may synergize with VEGF to promote pro-angiogenesis [117]. Gal-9 stimulates monocyte migration in vitro and produces acute inflammatory arthritis in mice, suggesting a unique role for gal-9 in angiogenesis, joint inflammation and other inflammatory diseases [118,119]. Reduced gal-9 and VEGF-A concentrations in women with previous miscarriages may be associated with angiogenic regulation [120]. Gal-12 has been shown to have an angiogenic effect in adipose tissue [121].

During pregnancy, gal-13 stimulates the dilatation of uterine arteries and veins via endothelium-dependent endothelial NO synthase (eNOS) and prostaglandin signaling pathways [89]. The carbohydrate recognition domain of gal-13 causes the structural stability of vasodilation by binding to the sugar residues of extracellular and connective tissue components [89].

## 5. Galectins in Pregnancy Disorders

The function of galectin dysregulation in abnormal pregnancies is becoming the focal point of investigation for an increasing number of researchers, with some issues linked to faulty placental development, abnormal angiogenesis and others with inflammatory responses and inappropriate maternal–fetal immunological tolerance. Figure 3 summarizes the expression of galectins in these four categories of pregnancy-related disorders and the possible pathogenic processes.

### 5.1. Preeclampsia

Preeclampsia is a multisystemic disorder specific to pregnancy, mainly associated with systemic small vessel spasms [122]. In early pregnancy, immune, genetic and endothelial cell dysfunction factors can lead to the spasm of the small spiral arteries of the uterus and a reduced invasiveness of trophoblast cells due to ischemia [123]. It has also been suggested that PE is associated with trophoblast immaturity, which has poor trophoblast differentiation in pathologies [124]. In mid to late pregnancy, due to local oxidative stress in the placenta from ischemia and hypoxia, endothelial cell damage is induced, resulting in the release of a large number of inflammatory factors, thus causing various clinical symptoms such as preeclampsia and eclampsia. The main clinical manifestations are hypertension and kidney damage [123].

PE is classified as early onset (before 34 weeks of gestation) or late onset based on its onset timing (after 34 weeks of gestation) [125]. Compared with the normal group, researchers found that gal-1 was elevated in early-onset and late-onset HELLP patients [126]. The term HELLP is used to describe a clinical disease that produces hemolysis, elevated liver enzymes and low platelet count. As a serious complication of hypertension during pregnancy, it can be fatal for both mother and fetus [127]. Low gal-1 levels at 18–24 weeks, but not 27–31 weeks, predicted early-onset and late-onset PE, as well as gestational hypertension (GH). After adjusting for the effects of high blood pressure and an elevated soluble fms-like tyrosine kinase-1 (sFlt-1)/placental growth factor (PlGF) ratio at 18–24 weeks, decreased gal-1 expression is considered to be a risk factor for the development of GH and PE in pregnancy [128]. Because it is expressed at low levels in late pregnancy and at high levels after the onset of the condition, serum gal-1 levels have been proposed as an independent risk factor for gestational hypertension and preeclampsia [128]. This is due to the fact that it is expressed at low levels in late pregnancy and at high levels following the onset of the condition [128].

According to the findings of Freitag et al., patients who presented with early-onset PE showed a reduced expression of gal-1 [106]. Galectin-1 suppresses trophoblast cell growth and promotes the development of syncytium. Its suppression in the syncytiotrophoblast has been linked to early pregnancy loss [129]. Recombinant Gal-1 also promotes differentiation and invasion of trophoblast stem cells [130]. These findings provide credence to the hypothesis that gal-1 is necessary for a healthy pregnancy and highlight gal-1 as a biomarker that has the potential to be helpful in the early diagnosis of PE [106].

Gal-2 is thought to be an inhibitor of atherogenesis by regulating monocyte/macrophage numbers in a mice model [115]. In PE, both spiral artery formation and macrophage inward flow were dramatically altered. As a result, abnormal spiral artery transformation in PE may be connected to trophoblast downregulation of gal-2 expression [131]. In preeclampsia, both mRNA and protein levels of gal-2 are decreased in EVT, independent of the time of PE onset. This discovery was made possible by the fact that gal-2 was shown to be one of the molecules with reduced expression [131]. A recent study found that downregulated gal-2 seems to prevent the apoptosis of Tregs in PE [132]. In trophoblasts, gal-2 has been linked to the histone modifications H3K4me3 (trimethylated lysine 4 of the histone H3) and H3K9ac (acetylated lysine 9 of the histone H3). In addition, an increase in syncytialization was seen following incubation with Gal-2 [133]. According to Charkiewicz et al. gal-2 participates in the immunological pathogenic process of PE [134]. The reduced gal-2 may be associated with autoantibodies against this protein along a potential immunological pathway. Individuals with antiphospholipid syndrome produce autoantibodies against gal-2 [135]. Antiphospholipid syndrome is prevalent in individuals with PE, which may help to explain why women who have mild forms of PE have lower levels of gal-2 in their blood [136].

During mouse pregnancy, gal-3 deficiency leads to placental dysfunction, which is manifested by inflammation and poor perfusion. This may be associated with an elevated expression of gal-3 in the EVT, leading to the development of PE [137,138]. Furthermore, the gal-3 levels of patients with late-onset PE were considerably lower than those of patients with early-onset PE [139].

Gal-7 increased anti-angiogenic sFlt-1 splice variants in the placenta and reduced the production and secretion of ADAM12 (a catabolite and metalloenzyme 12) and angiotensinogen, which may lead to the development of PE [140]. Tim-3/Gal-9 causes a pro-inflammatory response by regulating the polarization of decidual macrophages, causing macrophages to polarize towards M1, resulting in an increased expression of pro-inflammatory factors, such as TNF-α and IL-1β, and a decreased expression of anti-inflammatory factors such as TGF-β and IL-10. It can also cause placental dysfunction impairment [83]. After the administration of the recombinant gal-9 (rGal-9) protein, researchers found that liver and kidney damage as well as maternal placental dysfunction were reversed [83]. The monocyte may be implicated in the pathogenesis of PE via the Tim-3/Gal-9 pathway, and reducing Tim-3 may lower monocyte inhibitory activity [141]. The elevated expression of the Tim-3/Gal-9 pathway in PE may be involved in its systemic inflammatory response in PE, which shows that gal-9 may serve as a marker for PE [142].

Gal-13 is a lectin expressed by syncytiotrophoblast. The actin cytoskeleton, likely in conjunction with lipid rafts, regulates "nonclassical" PP13 exports from trophoblasts [143]. According to the results of studies, pregnant women with PE had lower levels of gal-13 than normal groups. However, there is no difference in gal-13 expression in the placenta or serum between pregnant women with PE and those who deliver at term [144]. In contrast, immune responses in STB microvilli were stronger in both preeclampsia and HELLP syndrome patients with term and preterm births than in controls [145]. In addition, gal-13 is highly expressed in syncytial cytoplasm protrusions, membrane vesicles and shed particles in PE and HELLP syndrome patients [143,145]. This mechanism may be due to the secretion of gal-13 into the intervillous space and perivenous area through the basal vein of the decidua, which forms gal-13 aggregates that attract and activate maternal immune cells and promote the transformation of trophoblast cells and small maternal spiral arteries [49]. PP13 induced uterine veins dilatation via the SKca–NO–BKca pathway in late pregnancy and became endothelium-dependent [146]. Researchers have proposed a gal-13 complementary therapy for the treatment of PE [89]. Screening serum gal-13 levels in the first trimester is a potential diagnostic method with high sensitivity and specificity for predicting preterm birth [147,148].

### 5.2. Fetal Growth Restriction

Fetal growth restriction (FGR), which can be diagnosed by ultrasound and maternal abdominal circumference measurements, is a disease in which the fetus does not grow to its proper potential in the uterus, usually due to placental hypoplasia and dysfunction [149,150]. A fetus with FGR is more susceptible to perinatal morbidity and mortality, as well as long-term health problems. Immediate complications include neonatal asphyxia, hypothermia and hypoglycemia, while long-term complications include intellectual disability, behavioral abnormalities and an increased incidence of hypertension, coronary heart disease and diabetes in adulthood [150,151]. Due to the importance of fetal growth for subsequent development and health and the absence of indicators for early identification and appropriate care, there is a need for in-depth study on this pregnancy issue and its prognosis [152].

Gal-1 expression is low in the serum and placenta of FGR-affected pregnant women. In addition, gal-1 may contribute to the development of FGR and serve as a possible diagnostic marker for the disease [153]. By immunohistochemistry, Jeschke et al. discovered a considerable elevation of gal-1 and gal-3 expression on EVT in placentas from patients with preeclampsia or HELLP, but no significant change was seen in FGR placentas compared to normal controls [137].

In the FGR, gal-2 and gal-13 expression was decreased in placental villous and extravillous trophoblasts; furthermore, this decrease was more pronounced in male placentas, which demonstrates significant sex differences [154]. In contrast, gal-3, the only chimeric lectin, showed no sex differences and was only significantly downregulated in extravillous cytotrophoblast of the placenta [154]. Decreased gal-3 expression during pregnancy in mice without preeclampsia syndrome leads to placental dysfunction, as well as FGR. This suggests that gal-3 deficiency leads to placental inflammation and malperfusion. Gal-3 dysregulation leads to abnormal uterine natural killer cell activation and infiltration, further contributing to the development of FGR [138]. These same findings were also observed in human pregnancies, where reduced maternal serum gal-3 levels were associated with the development of FGR in mid and late pregnancy [138].

Decreased gal-13 levels in early pregnancy are associated with a number of pathological pregnancy disorders, in particular fetal growth restriction and early onset preeclampsia. In the second and third trimesters, pregnant women with PE and FGR had concentrations that were greater than normal [155], mostlikely due to its strong association with placental growth before the transition of uterine spiral arteries begins [156]. One study, however, revealed no link between decreasing gal-13 levels and FGR. Further research is required to determine whether gal-13 assays have any relevance for assessing early pregnancies [157].

The expression of certain galectins is believed to correlate with the fetal gender. Gal-4, gal-8 and gal-9 expression in male fetal FGR trophoblast cells is dramatically reduced. In contrast, gal-9 and gal-12 expression increased in the EVT and endothelial cells of female fetuses with FGR [158]. Thus, researchers argue that tandem repeat galectins in FGR placentas have a fetal sex-specific role [158]. Using double immunofluorescence with trophoblast-specific markers, cells expressing galectin at the maternal–fetal interface in the decidua were identified. Only the extravillous trophoblast was significantly downregulated for gal-3 in FGR placentas. In contrast, both the villous and extravillous trophoblast cells of FGR placentas revealed lower gal-2 and gal-13 expressions [145,154].

### 5.3. Gestational Diabetes Mellitus

Gestational diabetes mellitus (GDM) is a serious pregnancy complication that includes two conditions: a pregnancy in a patient with pre-existing diabetes, known as pregestational diabetes mellitus (PGDM), and a first occurrence of diabetes after pregnancy, also known as gestational diabetes mellitus [159]. GDM is observed in about 90% of diabetic pregnant women. Changes to the diagnostic criteria for GDM have resulted in a considerable increase in the prevalence of GDM to above 15% [160]. Most patients with GDM recover from abnormal glucose metabolism after delivery; however, 60% will develop diabetes in the future [161]. GDM can lead to an increased incidence of spontaneous abortion in early pregnancy, complications of hypertension during pregnancy and a decrease in maternal resistance, which leads to infections. Effects on the fetus include fetal malformations and fetal growth restriction [162]. There is evidence that galectins contribute to the pathophysiology of GDM.

Elevated circulating gal-1 expression contributes to the development of GDM in mid- to late-term pregnancy, in addition to placental tissue gal-1 overexpression. Researchers detected aberrant gal-1 regulation in the local and peripheral circulation of the placenta in pregnancies with combined GDM. In addition, the relationship between LGALS1 polymorphism and GDM may suggest that genetic factors play a role in this unfavorable pregnancy outcome [163]. In GDM patients, there was an inverse relationship between glucose and gal-1, confirming these results [163].

A recent study revealed elevated gal-2 expression in STB and GDM placental decidua. These data suggest two possible conclusions about the function of gal-2 dysregulation in the pathophysiology of GDM: that elevated gal-2 expression is a reaction to the inflammatory state of GDM or whether it promotes the development of GDM [36]. Consequently, its significance in GDM and potential treatment implications must be further studied.

Gal-3 levels increased when mothers presented with gestational diabetes [164,165,166,167]. Women with late-onset GDM exhibited raised gal-3 levels in the first trimester, suggesting that gal-3 in the first trimester can serve as a predictor of GDM and may be related with increased insulin resistance [166,168]. Other researchers found that GDM patients’ circulating gal-3 levels were significantly lower than those of normal pregnant women [41]. As mid- to late-term placental development correlates with increased maternal peripheral gal-3 levels, it has been postulated that trophoblast cells may be a source of circulating lectins. In the same study, it was revealed that patients with GDM had reduced serum gal-3 levels, which were only detectable in late pregnancy. This may suggest that gal-3 is vulnerable to GDM-specific hormonal and metabolic changes [41].

Gal-4 overexpression in the decidua of women with type 2 diabetes, with significantly increased nuclear and cytoplasmic levels [169].

The gal-13 serum levels in the blood of GDM patients were considerably lower than those of healthy controls, as tested by ELISA. Since gal-13 has anti-inflammatory functions and regulates the maternal immune system, a deficiency of gal-13 may lead to an imbalance in the inflammatory process of the placenta during pregnancy, thereby influencing the development of GDM [170].

### 5.4. Preterm Birth

Preterm birth is defined as less than 37 weeks of gestation [171]. The shorter the weeks of gestation at delivery, the lower the birth weight, the worse the prognosis for the perinatal baby and the more complications after birth, accounting for 5% to 18% of pregnancy complications [172]. The causes of the majority of PTBs, which are spontaneous and idiopathic, are largely unknown [172].

Preterm premature rupture of the membranes (PPROM) refers to the rupture of the fetal membranes prior to the 37th week of pregnancy. This problem is estimated to affect 4% of pregnant women and accounts for nearly half of all preterm births [173]. Despite significant improvements in perinatal care over the past few decades, the number of preterm births caused by PPROM continues to increase [174]. It is hypothesized that galectins are related to premature membrane rupture and that premature membrane rupture is associated with up to 75% bacterial membrane and/or amniotic fluid infections [175]. Compared with healthy controls, PPROM patients showed significantly higher levels of gal-1 and gal-3. Gal-3 concentrations in maternal serum are significantly and adversely linked with gestational age and birth weight. The data show that gal-1 and gal-3 regulate critical biological processes and may be initiators of PPROM pathogenesis, predictive indicators, and targets for preventative treatments [176]. Chorioamnionitis is an acute inflammatory reaction associated with early rupture of the amniotic sac. Inflammatory factors cause chorioamnionitis by disrupting membrane junctions [177]. Gal-1 is associated with an inflammatory response [178]. IL-1, a key cytokine in chorioamnionitis, can upregulate gal-1 expression [179]. Overexpression of gal-1 in fetal membranes may correlate with weakened membrane structure, increased susceptibility to infection and eventual membrane rupture, and it is believed that gal-1 protects the fetus against bacterial infection [180]. At 21–24 weeks of gestation, the majority of spontaneous births are accompanied by histological chorioamnionitis, whereas at 35–36 weeks, this rate is approximately 10% [181]. Than et al. revealed that gal-1 modulates the immunological response to infection and inflammation and that chorioamnionitis is related with high gal-1 mRNA expression and robust inflammatory immunoreactivity. Therefore, gal-1 may be implicated in the modulation of the inflammatory response of chorioamnionitis [180].

The immunological regulator Gal-3 was highly increased in placenta, amniotic fluid and serum. Co-stimulation with gal-3 and Porphyromonas gingivalis (*P.g.*)-LPS raised cytokine levels; however, co-stimulation with gal-3 and Aggregatibacter actinomycetemcomitans (*A.a.*)- or Escherichia coli (*E. coli*)-LPS lowered cytokine levels, showing the essential involvement of gal-3 in P.g.-induced PTB. Infection with P.g.-dental produced PTB, which was related with gal-3-dependent cytokine production, prompting the development of gal-3 targeted therapy or diagnostic systems for PTB treatment [73]. A link between gestational age and gal-3 levels was identified by analyzing gal-3 levels in the cord blood of term and preterm newborns. Expression of gal-3 was induced by invasive but not definitive streptococcus lactis strains, indicating a role for gal-3 in innate immunity [182]. Nevertheless, studies found the gal-3 binding protein levels in cord blood have revealed that gal-3BP levels are elevated in cases of preterm birth, which may reflect the inflammatory status of the infant and mother, and this warrants further investigation [183].

A single facility conducted a prospective analysis of 170 preterm infants delivered at fewer than 35 weeks gestation. During the neonatal period, peripheral venous blood was collected, and gal-1, gal-3 and gal-9 were measured by ELISA [184]. Researchers found that negative correlation existed between the levels of gal-1 and gal-3 at delivery and gestational age. Gal-1 and gal-9 concentrations were significantly higher in neonates with Amniotic Infection Syndrome (AIS). Neonates with early-onset sepsis exhibited greater gal-3 concentrations than healthy infants. Observational research revealed that preterm children born in an inflammatory milieu, such as AIS or early-onset sepsis (EOS), have higher levels of galectin-1, -3 and -9. Future studies must identify if galectins mediate inflammation-induced preterm birth; if so, galectins might be the subject of treatment studies [184].

## 6. Conclusions

Galectins are a family of conserved, soluble proteins distinguished by their affinity for the galactoside structures present on a variety of glycoconjugates. Over the past several decades, galectins have been recognized as crucial for implantation and pregnancy maintenance. The results of several studies have revealed their role in trophoblast cell function and placental development. In addition, evidence suggest that they play key roles in the control of fetal–maternal immunological tolerance and angiogenesis. Additionally, an increasing number of studies reported pregnancy-related diseases with altered or dysregulated galectin expression.

This review seeks to elucidate the expression of the galectin family in the reproductive system and its function in normal and pathological pregnancy. An increasing number of researchers haves found galectins, and a substantial body of this literature indicates that specific galactose lectins may be recommended to predict pregnancy-related diseases. However, additional research is required to confirm this.

## Figures and Tables

**Figure 1 ijms-23-10110-f001:**
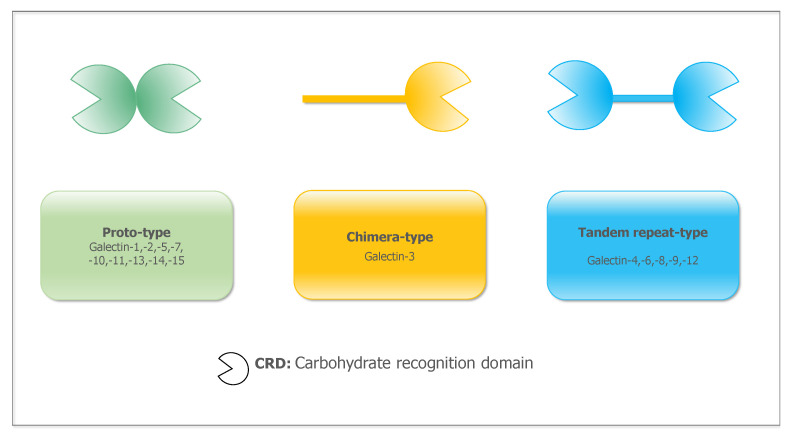
The classification and structure of galectins. The proto-type galectins are in green, which are non-covalently connected. The chimera-type galectins are in yellow, which are self-associated with a c-terminal CRD and a non-carbohydrate bound n-terminal structural domain. The tandem repeat-type galectins are in blue, which are dimers consisting of a linker peptide joining two CRDs.

**Figure 2 ijms-23-10110-f002:**
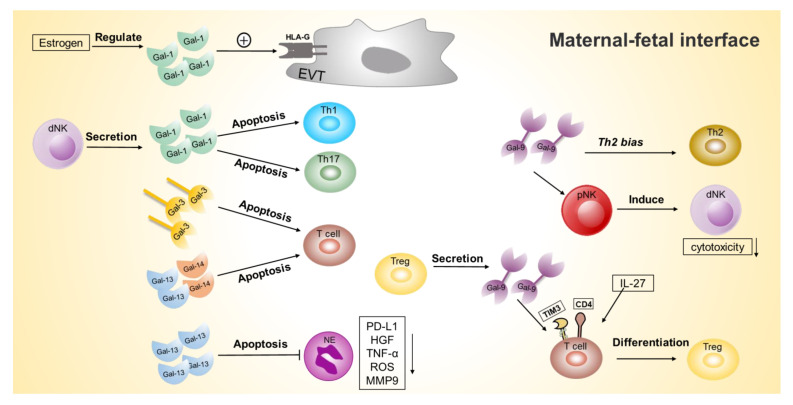
Galectin affects the function of immune cells at the maternal–fetal interface, maintaining maternal–fetal immunological tolerance. (1) Gal-1, regulated by estrogen, modulates HLA-G expression on EVT. (2) dNK cells produce gal-1, which can induce apoptosis of Th1 and Th17 cells. (3) Gal-3 induces the apoptosis of T cells. (4) Gal-13 and gal-14 can induce the apoptosis of T cells. (5) Gal-13 reduces the rate of apoptosis in neutrophils and increases the expression of PD-L1 and the production of HGF, TNF-α, ROS and MMP-9 in neutrophils. (6) Gal-9 signal is important for the regulation of PBMC function toward a Th2 bias. (7) Gal-9 induce peripheral NK cells to a dNK-like phenotype. (8) Treg cells have high levels of gal-9, and gal-9 interacts with Tim-3 to promote the differentiation of decidual Tim-3+ CD4+ T cells into Treg cells.

**Figure 3 ijms-23-10110-f003:**
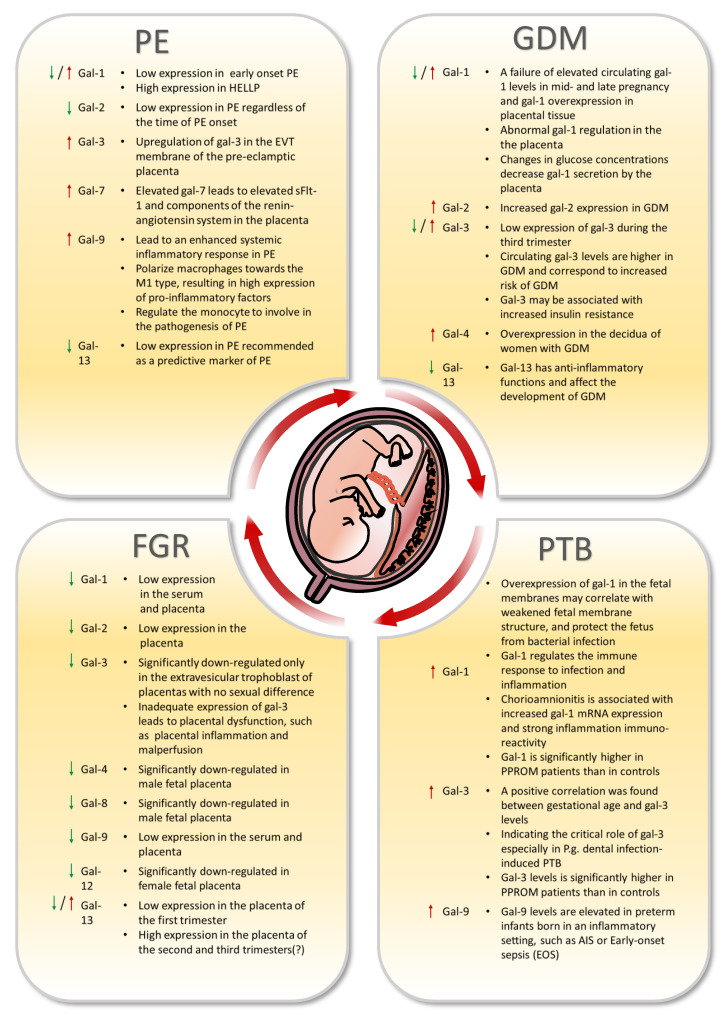
Expression of galectins in four types of pregnancy-related diseases and their possible pathological mechanisms. The four pregnancy-related diseases are preeclampsia (PE), gestational diabetes mellitus (GDM), fetal growth restriction (FGR) and preterm birth (PTB). The figure describes the dysregulation of galectins in these diseases. Red up-arrow means up regulation; Green down-arrow means down regulation.

**Table 1 ijms-23-10110-t001:** The expression of galectins in maternal–fetal interface.

Galectins	Expression	Reference
Galectin-1	Decidua; Endometrium; Extravillous trophoblast Immune cells (i.e., dNK; CD4 + CD25 + Treg cells); Syncytiotrophoblast	[29,47,52,53]
Galectin-2	Decidua; Endometrium; Extravillous trophoblast; Syncytiotrophoblast	[29,38]
Galectin-3	Cytotrophoblast; Endometrium; Villous trophoblast	[29,39,40]
Galectin-4	Endometrium	[29]
Galectin-7	Decidua; Endometrium; Extravillous trophoblast; Glandular epithelial cells; Syncytiotrophoblast	[29,42,54]
Galectin-8	Endometrium; Syncytiotrophoblast; Extravillous trophoblast; Villous trophoblast	[29,43]
Galectin-9	Cytotrophoblast; Decidua; Endometrium	[29,44]
Galectin-10	Immune cells (i.e., CD4 + CD25 + Treg cells)	[47]
Galectin-12	Endometrium	[29]
Galectin-13	Syncytiotrophoblast	[49,50]
Galectin-14	Syncytiotrophoblast	[7]
Galectin-15	Caprinae endometrium	[51]

## Data Availability

Not applicable.

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
