# Peer review of "Galectins: Important Regulators in Normal and Pathologic Pregnancies"

_ijms, 2022, doi:10.3390/ijms231710110_

Round 1

Reviewer 1 Report

This is an interesting review on the role of galectins during pregnancy. Nevertheless, the manuscript needs deep revision to improve its scientific quality. Moreover, authors must correct the grammatical and spelling errors.

In the following lines I highlight some specific modifications that the authors must address:

Lines 53-54. “the surrounding tissue of the cell” What do the authors refer? The extracellular matrix? If so, please use appropriate terminology.

Line 56. “they utilize nonclassical secretion pathways”. This sentence is too ambiguous. Authors must specify which are these “nonclassical” secretion pathways.

Lines 63-64. “Extracellular galectins are capable of binding to various cell surface receptors to form carbohydrates”. Authors must revise this sentence and provide a clear message.

Line 90. “Gal-3 was found by immunohistochemical”. Please, revise and improve this sentence.

Line 211. “The process of a successful embryo implantation is a complicated one”. Please, revise and improve this sentence.

Lines 504-505. Please, write the scientific names in italics.

Line 523. Please change “Future study” by “Future studies”.

Conclusions: This section needs deep revision and improvement to clearly highlight the physiological relevance of galectins.

References 3 and 32. These references must be corrected.

The reference list only contains a publication of 2022. So, authors must update the references and include recent publications.

Reviewer 2 Report

This is a very interesting review resuming the current knoledges regarding the role of Galectins in pregnancy complications. Although the manuscript is well organised, some important points should be impoved before publication. In particular: 

Line 320: It deserves to be pointed out that PE is also characterized by trophoblast immaturity (PMID: 32529396). This is an important point to underline because it can contribute to Gals impairment found in this pathology (since, as reported by the authors, many Gals are also expressed in villous and extravillous cytotrophoblast).

Lines 493-496: this section is very interesting because little is known about Gals modulations in fetal membranes but it desearve to be added that in chorioamnionitis PPROM is mainly due to the action of inflammatory cytokines that lead to the disruption of cell junctions of foetal membranes making them prone to breakage (PMID: 26739007). This is an important process in chorioamnionitis and preterm birth since the same process could be the cause of Gals modulation in this pregnancy complication. In fact, IL-1 (a key cytokine in chorioamnionitis) can upregulate Gal 1 expression (PMID: 24503185) further validating the studies proposed by the authors.

An accurate revision of punctuation and syntax is recommended

Round 2

Reviewer 1 Report

Authors have introduced all the modifications requested and the overall quality of the manuscript has been improved. Nevertheless, the text still contains grammatical and typographical errors that must be corrected.

Reviewer 2 Report

the manuscript can be accepted in the present form.